


# Year-round record of bulk and size-segregated aerosol composition in central Antarctica (Concordia site) Part 2: Biogenic sulfur (sulfate and methanesulfonate) aerosol

Michel Legrand[1,2], Susanne Preunkert[1,2], Rolf Weller[3], Lars Zipf[4], Christoph Elsässer[4], Silke Merchel[5],
Georg Rugel[5], and Dietmar Wagenbach[4,*]

[1]Université Grenoble Alpes, Laboratoire de Glaciologie et Géophysique de l'Environnement (LGGE), Grenoble, 38402, France
[2]CNRS, Laboratoire de Glaciologie et Géophysique de l'Environnement (LGGE), Grenoble, 38402, France
[3]Alfred Wegener Institut für Polar und Meeresforschung, Bremerhaven, 27570, Germany
[4]Institut für Umweltphysik, University of Heidelberg, Heidelberg, 69120, Germany
[5]Helmholtz-Zentrum Dresden-Rossendorf (HZDR), Dresden, 01328, Germany
[*]Deceased December 2014

Correspondance to M. Legrand (Michel.Legrand@univ-grenoble-alpes.fr)

**Abstract.** Multiple year-round (2006-2015) records of the bulk and size-segregated composition of aerosol were obtained at the inland site of Concordia located in East Antarctica. The well-marked maximum of non-sea-salt sulfate ($nssSO_4$) in January ($84 \pm 25$ ng m$^{-3}$ against $4.4 \pm 2.3$ ng m$^{-3}$ in July) is consistent with observations made at the coast ($280 \pm 78$ ng m$^{-3}$ in January against $16 \pm 9$ ng m$^{-3}$ in July at Dumont d'Urville, for instance). In contrast, the well-marked maximum of MSA at the coast in January ($60 \pm 23$ ng m$^{-3}$ at Dumont d'Urville) is not observed at Concordia ($4.6 \pm 2.4$ ng m$^{-3}$ in January). Instead, the MSA level at Concordia peaks in October ($5.6 \pm 1.9$ ng m$^{-3}$) and March ($13.2 \pm 6.1$ ng m$^{-3}$). As a result, a surprisingly low MSA to $nssSO_4$ ratio ($R_{MSA}$) is observed at Concordia in mid-summer ($0.05 \pm 0.02$ in January against $0.25 \pm 0.09$ in March). We find that the low value of $R_{MSA}$ in mid-summer at Concordia is mainly driven by a drop of MSA levels that takes place in submicron aerosol (0.3 μm diameter). The drop of MSA coincides with periods of high photochemical activity as indicated by high ozone levels, strongly suggesting the occurrence of an efficient chemical destruction of MSA over the Antarctic plateau in mid-summer. The relationship between MSA and $nssSO_4$ levels is examined separately for each season and indicates that concentration of non-biogenic sulfate over the Antarctic plateau does not exceed 1 ng m$^{-3}$ in fall and winter and remains below 5 ng m$^{-3}$ in spring. This weak non-biogenic sulfate level is discussed in the light of radionuclides ($^{210}$Pb, $^{10}$Be, and $^{7}$Be) also measured on bulk aerosol samples collected at Concordia. The findings highlight the complexity in using MSA in deep ice cores extracted from inland Antarctica as a proxy of past DMS emissions from the southern ocean.

Keywords: Methanesulfonate, MSA to non-sea-salt sulfate ratio, DMS emissions, sea-salt aerosol, $^{210}$Pb, $^{10}$Be and $^{7}$Be, Chemistry (chemical composition and reactions)



## 1. Introduction

The coupling between climate and atmospheric aerosol involves complex processes that are not yet fully elucidated. In the southern hemisphere, aside from the primarily emitted sea-salt particles, the oxidation of dimethyl sulfide (DMS) emitted by phytoplankton is an important source of secondary aerosol (Gondwe et al., 2003). In the atmosphere, DMS is oxidized into

small sulfate and methanesulfonate aerosols that interact with solar radiations reaching Earth's surface by scattering of solar energy and by acting as condensation nuclei for cloud droplets, thereby affecting the cloud albedo (Shaw, 1983; Charlson et al., 1987).

Polar ice cores provide a unique archive of climate and past atmospheric aerosol (composition and load) that may help to address some relevant key questions (Legrand and Mayewski, 1997). In contrast to sulfate, methanesulfonate ($MS^-$, also

denoted MSA) is exclusively formed by photo-oxidation of DMS. Pioneering studies dedicated to its records extracted from Antarctic ice cores proposed its use to investigate changes of the marine biota in response to past climatic fluctuations (Legrand and Feniet-Saigne, 1991; Legrand et al., 1991) or sea-ice extend (Welch et al., 1993; Curran et al., 2003). However, it rapidly appears that the interpretation of MSA ice core profiles in terms of past oceanic DMS emissions is far less straightforward than initially thought. First, highly complex mechanisms control the DMS marine emissions. For instance, it

is now recognized that the concentration and the oceanic emission of DMS is not only controlled by the phytoplankton biomass or activity alone, but also by numerous not well understood ecological and biogeochemical processes (Simó and Dachs, 2002). Second, the atmospheric behaviour of DMS which is characterized by a variable MSA oxidation yields (Gondwe et al., 2004) renders more difficult than expected the use of the MSA to non-sea-salt sulfate ($R_{MSA}$) ratio to separate the contribution of marine biogenic emissions from other sulfate sources like volcanic activity, terrestrial sources, and

possible the stratospheric sulfate reservoir. It is now well recognized that $R_{MSA}$ is highest in polar region and lowest within the tropics, due to a more efficient MSA production from the OH oxidation of DMS at low temperatures (Bates et al., 1992; Gondwe et al., 2004). However, at very high latitudes the atmospheric behaviour of DMS may be even more complex than elsewhere due to the presence of halogenated radicals (Read et al., 2008) and a possible role of heterogeneous chemistry on the behaviour of DMSO (Davis et al., 1998). Whereas a good relationship between the MSA level in air and fresh snow has

been observed (Jaffrezo et al., 1994; Wolff et al., 1998), the existence of post depositional losses and migration of MSA signals within annual firn layers has been recognized by several studies (Wagnon et al., 1999; Pasteur and Mulvaney, 2000; Delmas et al., 2003; Weller et al., 2004). Concerning the loss of MSA from the Antarctic snowpack towards the atmosphere, the subsequent presence of MSA in the gas phase is however still unclear (Weller et al., 2004; Piel et al., 2006; Mauldin et al., 2004). This loss of MSA was proposed to explain the previous observations of a decreasing trend of $R_{MSA}$ in snow

deposited at the coast compared to inland Antarctica (Legrand, 1997). Finally, the calculations of the non-sea-salt sulfate present in Antarctica is less easier than at any other places of the world due to a depletion of sulfate relative to sodium caused by precipitation of mirabilite ($Na_2SO_4.10\ H_2O$) during freezing of seawater in winter (Wagenbach et al., 1998).





To progress on these questions, atmospheric records of both DMS and sulfur aerosol are needed particularly in the vicinity of sites where Antarctic ice cores are extracted. While detailed long-term records of sulfur derived aerosol species (sometimes completed by DMS and DMSO measurements) are available for the coastal sites of Neumayer and Dumont d'Urville (Wagenbach, 1996; Minikin et al., 1998; Jourdain and Legrand, 2001), only very scattered atmospheric observations of both

MSA and sulfate are obtained so far at central Antarctic positions. Except the study conducted at Concordia by Preunkert et al. (2008) covering a complete annual cycle of MSA, non-sea-salt sulfate and DMS, most of inland records were restricted to austral summer with only few data obtained during polar night (Arimoto et al., 2001, 2004, and 2008; Udisti et al., 2004; Piel et al., 2006).

We here report on multiple year-round (2006-2015) records of bulk aerosol composition of sulfur-derived aerosol (MSA and

sulfate) at the Concordia site located on the high East Antarctic plateau. The record of bulk aerosol is complemented by a study of the size-segregated aerosol composition conducted by running a 12-stage impactor over three years (2009-2011). As discussed in the companion paper (Legrand et al., this issue), these impactor data are essential to evaluate the degree of sulfate depletion relative to sodium of sea-salt aerosol in winter and consequently to accurately calculate here the nss-$SO_4$ level and the $R_{MSA}$ ratio. Here we present and discuss the temporal variability of the composition of sulfur-derived aerosol

(MSA, non-sea-salt sulfate, and $R_{MSA}$) and its dependence to the aerosol-size, in relation with seasonal change of marine biogenic DMS emissions and the contribution of non-biogenic sources of sulfate.

## 2. Sites, Samplings, and Methods

Bulk aerosol sampling was initiated in 2006 at the inland site of Concordia (75°06'S, 123°20'E, 3233 m asl) located near Dome C (DC, 1100 km away from the nearest coast of East Antarctica). Working conditions are detailed in Legrand et al.

(this issue). Given the weekly sampling time, a large air volume was sampled ($\sim$ 8000 m$^3$) permitting the blank values to remain well below 1 ng m$^{-3}$ (0.17 ± 0.15 ng m$^{-3}$ for sodium, 0.4 ± 0.3 ng m$^{-3}$ for sulfate, and zero for MSA). As reported in Fig. 1, bulk aerosol chemical measurements were backed up by measurements of the $^{210}$Pb activities (310 samples) using γ-spectrometric quantification, as detailed by Wagenbach et al. (1988) and Elsässer et al. (2011). Due to the short half life (53 days) of $^7$Be, its γ-spectrometric quantification was mainly done on September-January samples, just after their retrograde in

Europe at the end of the austral summer season. In this way, $^7$Be measurements were obtained on 48 HV filters collected in summer 2006-2007, 2008-2009, 2009-2010, 2010-2011, and end of 2015, permitting to document the most important change of the $^7$Be/$^{210}$Pb activity ratio, as previously reported for the coastal Antarctic site of Neumayer (see Sect. 3.3.1). In addition to the non-destructive γ-spectrometry of $^{210}$Pb and $^7$Be, $^{10}$Be was chemically extracted from aliquot of filters collected in 2008 to allow quantification by accelerator mass spectrometry (AMS) at Dresden (DREAMS, Rugel et al., 2016). It is

important to emphasize here that, whereas $^{10}$Be is rather routinely measured in ice cores, its measurement in air samples is rather rare. Basic steps of chemical treatment are: (1) leaching of the filter with 10 mL very diluted HCl (ultrasonic bath for 5 min, resting over-night) in the presence of ~300 μg of $^9$Be-carrier (Scharlau, 2% HCl, $^9$Be concentration of 980.4 ± 4.9 μg





$g^{-1}$); (2) filtration through PVDF-filter (pore size of 0.45 μm); (3) precipitation of beryllium hydroxide by ammonia solution (25%); (4) rinsing three times with dilute ammonia solution (pH 8-9); (5) drying and ignition to BeO at 900°C; and (6) mixing with Nb-powder (1:6 by weight) and pressing into Cu cathodes. Every ninth sample was accompanied by a processing blank, which was treated identically as the filter samples. $^{10}$Be data and subsequent $^{10}$Be/$^{9}$Be AMS measurements

result in ratios of $2\ 10^{-13}$ to $1\ 10^{-11}$ with total uncertainties from 2.0-4.4% (mean uncertainty 2.4%).

The concentrations of nssSO$_4$ corresponding to HV samples were calculated as follows:

$$nssSO_4 = SO_4 - k_{SO4/Na}\,Na \qquad (1)$$

Examination of the size-segregated composition of aerosol present at Concordia indicates significant sulfate depletion relative to sodium with respect to the seawater composition from May to September (i.e. a $k_{SO4/Na}$ value of 0.16 ± 0.09

instead of 0.25 in seawater) (Legrand et al., this issue). This value of 0.16 was used in equation 1 to calculate the nssSO$_4$ concentrations. From November to April, an absence of sulfate depletion relative to sodium in sea-salt aerosol is assumed and a $k_{SO4/Na}$ value of 0.25 (i.e., the seawater reference value) was applied in equation 1.

The uncertainties in calculating the nssSO$_4$ level are related to the accuracy of determinations of SO$_4$ and Na as well as the uncertainties of the calculated value of $k_{SO4/Na}$:

$$\Delta(nssSO_4)^2 = (k_{SO4/Na}\,\Delta Na)^2 + (Na\,\Delta k_{SO4/Na})^2 + (\Delta SO_4)^2 \qquad (2)$$

With $k_{SO4/Na}$ equal to 0.25 in summer (November-April) and 0.16 ± 0.09 in winter (May-October), $\Delta SO_4^2 = (0.05\ SO_4)^2 + \sigma_{blank}^2$ and $\Delta Na^2 = (0.05\ Na)^2 + \sigma_{blank}^2$.

As discussed above, the HV blanks lead to a $\sigma_{blank}$ of 0.15 ng m$^{-3}$ for Na and 0.3 ng m$^{-3}$ for SO$_4$.

Uncertainties in calculating the MSA to nssSO$_4$ ($R_{MSA}$) ratio were estimated as follows:

$$\Delta R_{MSA}^2 = (\Delta MSA/nssSO_4)^2 + (MSA\,\Delta nssSO_4/nssSO_4^2)^2 \qquad (3)$$

with $\Delta MSA = 0.05\ MSA$

On a total of 446 HV filters, in 9 cases we calculate $R_{MSA}$ values that are out of order, corresponding to low nssSO$_4$ values (< 10 ng m$^{-3}$). These data (see the caption of Fig. 1) were not considered when calculating the monthly $R_{MSA}$ means reported in Fig. 2.

In addition, the size-segregated aerosol composition was investigated by doing 105 samplings between March 2006 and January 2012 by using a small deposit area impactor, equipped with a 20 μm cut-off diameter inlet (Legrand et al., this issue). Applying a sampling interval of 2 weeks, eight run per year were done in 2006 and 2007 and a more continuous sampling (25 runs per year) from 2009 to 2012. The blank values of the deposit remain well below 1 ng m$^{-3}$ (0.17 ± 0.12 ng m$^{-3}$ for sodium, 0.08 ± 0.06 ng m$^{-3}$ for sulfate, and zero for MSA). All data were blank corrected. The nssSO$_4$ and $R_{MSA}$ values were

calculated by applying in equation 1 a $k_{SO4/Na}$ value of 0.25 in summer (November-April). For winter sampling (from May to October), the calculations were done for each impactor run by using the individual $k_{SO4/Na}$ value derived from the corresponding impactor run, as detailed in Legrand et al. (this issue). Briefly, the $k_{SO4/Na}$ values are derived by examining the



levels of sulfate and sodium present on the stages where most of sea-salt aerosol was collected (0.5-2.0 μm diameter) and having corrected sulfate from its small (but significant) biogenic sulfate contribution, as estimated from MSA levels.

## 3. Results and discussions

As shown in Fig. 1, both $nssSO_4$ and $^{210}Pb$ levels in bulk aerosol collected at Concordia steadily increase from September to November and decrease from February to April. In Antarctica, the seasonal change of $^{210}Pb$, which is useful to trace the long-range transport of continental sub-micron aerosol, is characterized by summer maximum mainly driven by (1) strong inversion layer in winter particularly at inland sites, (2) seasonal change in the efficiency of the meridional long-range transport (Elsässer et al., 2011). At Concordia, the seasonal $^{210}Pb$ amplitude (a factor of 3 from May-August to November-February) is weaker than that of $nssSO_4$ (more than a factor of 10, Fig. 2). Since $^{210}Pb$ and $nssSO_4$ are both present in the atmosphere as submicron aerosol, this difference cannot be attributed to different atmospheric lifetime. Instead, this difference implies a strong seasonal change of sulfur emissions, particularly from September to November and February to April. That is supported by satellite observations showing that concentrations of chlorophyll at high latitudes in the southern surface ocean are increased and decreased at spring and fall equinoxes, respectively (Fig. 2).

In the following we discuss the respective abundance of the two sulfur species at Concordia, their seasonal cycle and variability over the 9 year-round records (Sect. 3.1). We then focus discussions on the striking weakness in the abundance of MSA with respect to $nssSO_4$ observed during mid-summer over inland Antarctica (Sect. 3.2). Finally, in Sect. 3.3 we examine the importance of non-biogenic sources of sulfate for inland Antarctica over the course of the year.

### 3.1 Seasonal cycle of MSA and NssSO₄ levels at Concordia

In winter (June-September), the levels of MSA and $nssSO_4$ at Concordia remain as low as $0.6 \pm 0.4$ ng m$^{-3}$ and $6.4 \pm 2.2$ ng m$^{-3}$, respectively (Table 1). The relative contribution of marine biogenic versus non-biogenic sources to the sulfate budget over the Antarctic plateau in winter will be discussed in Sect. 3.3. From winter to November, MSA and $nssSO_4$ levels exhibit a similar increase by a factor of 9 to 10 (Table 1). This increase is larger than the ones seen at coastal sites (a factor of 6 to 7 for MSA and close to a factor of 5 for $nssSO_4$, Table 1). The increase of MSA and $nssSO_4$ from September to November at the coast were attributed by Minikin et al. (1998) to the recovery of the marine biota in the southern ocean. The larger increase of the two sulfur species from winter to spring at Concordia compared to coastal sites is likely related to weakening of the inversion layer at Concordia at the end of winter that also contributes to the increase there. As seen in Table 1, both at coastal sites and Concordia the $R_{MSA}$ ratio remains close to 0.08 in winter. Such a low value of $R_{MSA}$ is discussed in terms of source region of biogenic sulfur in Sect. 3.3.2.

The maximum of $nssSO_4$ in January seen in multiple year records available at the coastal sites of Neumayer (NM) and Dumont d'Urville (DDU) (Fig. 3) is also observed in the multiple-year $nssSO_4$ record at Concordia (Fig. 2). As seen in Table 1, the $nssSO_4$ levels are coastal sites are enhanced by around a factor of two from November to January. At Concordia, the



nssSO$_4$ levels are consistently increasing from $63.6 \pm 22.5$ ng m$^{-3}$ in November to $84.4 \pm 25.3$ ng m$^{-3}$ in January. A larger increase of MSA compared to the nssSO$_4$ is observed from November to January at the coast (more than a factor of 3, Table 1) leading to a large increase of R$_{MSA}$. Given the latitudinal dependence of R$_{MSA}$ characterized by high values at very high latitudes (>60°S), such a difference in the recovery of MSA and nssSO$_4$ in January is expected since the activity of marine

biota at latitudes higher than 60°S is peaking at that time (Fig. 2). Note the lower value of R$_{MSA}$ in January at DDU compared to NM (Table 1) that will be discussed in Sect. 4.

At Concordia, a quite different picture emerges for MSA with levels peaking before and after sulfate (in November and March, Fig. 2). Even more dramatic is the difference of R$_{MSA}$ between Concordia and coastal sites with values dropping at 0.05 in January at Concordia (Fig. 2) against around 0.2 at DDU and 0.4 at NM (Fig. 3). Given the relative abundance of

MSA with respect to nssSO$_4$ at the coast, with a mean level of 84 ng m$^{-3}$ of nssSO$_4$ at Concordia in January we would expect there between 17 and 34 ng m$^{-3}$ of MSA at that time (i.e., 4 to 7 times higher than the mean observed level of 4.7 ng m$^{-3}$, Table 1).

A few previous studies already pointed out the occurrence of R$_{MSA}$ as low as 0.1 or less at inland Antarctic sites in mid-summer. As seen in Table 2, most of observations were restricted to a few weeks in in December and/or January, except the

one conducted at Kohnen (located at 2890 m asl in Dronning Maud Land) by Weller and Wagenbach (2007) where a composite annual cycle based on discontinuous sampling done over 2.5 years was obtained. In addition, Preunkert et al. (2008) reported continuous samplings done at Concordia over the 2006 year. However none of them examined in detail the change of R$_{MSA}$ over the course of summer (from spring to summer and fall) and its variability from year to year.

### 3.2. Causes of the weak abundance of MSA compared to nssSO$_4$ in mid-summer

### 3.2.1. Previous invoked causes of low R$_{MSA}$ in summer at inland Antarctica

Several aspects have to be considered in discussing causes of the surprising decrease of R$_{MSA}$ in December/January compared to values in October and March at Concordia, contrasting with what is observed at the coast. They mainly include segregation between MSA and sulfate during transport towards Dome C via either formation and/or deposition of the two species. In addition, as proposed to explain the loss of MSA from the snowpack, we cannot exclude the possibility of an

evaporative loss from aerosol.

Here we may first invoke a different size distribution of nssSO$_4$ and MSA leading to a change of their respective abundance during transport between the ocean and central Antarctica. Indeed, several studies (Rankin and Wolff, 2003; Kerminen et al., 2000) pointed out an enrichment of MSA with respect to nssSO$_4$ in micron compared to submicron particles at coastal Antarctic sites in summer. In this way, the decrease of R$_{MSA}$ at Concordia in December and January could be explained by

(1) a larger abundance of MSA with respect to nssSO$_4$ in large than in small particles at the coast and (2) a transport of marine air mass between the coast and Concordia becoming less efficient in December and January compared to October and March. An alternative possibility involves a selective formation of sulfate with respect to MSA under mid-summer





conditions inland Antarctica. As discussed by Davis et al. (1998), the $R_{MSA}$ ratio in Antarctica is strongly influenced by the respective importance of liquid and gas phase sulfur chemistry. Briefly, the OH oxidation of DMS produces $SO_2$ (abstraction pathway), dimethyl sulfoxide (DMSO) and $SO_2$ (addition pathway). DMSO is further oxidized by OH either in the gas phase or in the aqueous or aerosol phase. Legrand et al. (2001) reported that the heterogeneous DMSO oxidation produces

efficiently MSA in summer at the coast. The efficiency of this heterogeneous process has been confirmed by kinetic studies (Bardouki et al., 2002). Therefore, due to a larger presence of aerosol, it is expected that the oxidation of DMSO would produce much more MSA in the atmospheric boundary layer compared to the buffer layer above. That was supported by field observations made by Davis et al. (1998) at the Palmer site showing a rapid increase of DMSO when vertical downward transport brought buffer layer air mass within the boundary layer. These observations imply a longer lifetime of

DMSO in the buffer layer than in the boundary layer, following a strong weakening of the heterogeneous reaction of DMSO caused by a far lower aerosol surface (and liquid water) available there. From that, Preunkert et al. (2008) proposed that the weakening of marine advection in December/January compared to March associated with an on-going oxidation of $SO_2$ into sulfate in the buffer layer, where the heterogeneous chemistry of DMSO is very limited, would account for the drop of $R_{MSA}$ observed in mid-summer at Concordia. Note however that this hypothesis was based on observations that were limited to one

15   year.

### 3.2.2. Size-segregated composition of sulfur aerosol at Concordia in summer

The previously mentioned enrichment of MSA with respect to $nssSO_4$ seen in micron compared to submicron particles at the coast in summer is also observed at DDU, a coastal site located in margin regions facing the Indian and Pacific oceanic sectors from where are coming most of marine air mass reaching Concordia. As shown in Fig. 4, the second micron mode of

MSA in aerosol at DDU remains centred around 1-2 μm, and differs from the coarse sea-salt aerosol mode observed at 6-8 μm by Jourdain and Legrand (2001 and 2002). The contribution of the second mode to the total MSA mass remains rather weak (less than 18%). At Concordia in summer, the size distribution is very similar to the one observed at DDU with no significant decreasing contribution of the micron particles to total MSA mass between the two sites (Fig. 4). Furthermore, as seen in Fig. 5, the size-distributions of the two sulfur species typically observed at Concordia indicate no significant

decreasing contribution of the micron mode to the total MSA mass in summer compared to other seasons (16% in winter, 18% in spring and summer, and 10% in fall). That suggests an absence of selective deposition of MSA during transport between the coast and the inland Antarctic plateau in mid-summer. The weak contribution of micron particles to the total MSA mass, as observed at the coast, likely limits the segregation between MSA and $nssSO_4$ during the transport between ocean and Concordia. It is also interesting to notice that the dominant presence of MSA in submicron sulfuric acid particles

does not confirm the statement generally presented in previous studies that, after its formation in the gas phase, in contrast to sufuric acid, MSA is more easily incorporated in larger less acidic particles (Jefferson et al., 1998). Since DMSO is soluble in acidic solution, we may in fact expect that its solubilisation in hydrated acidic aerosol followed by a rapid oxidation into MSA explains the presence of this latter in sulfuric acid particles.

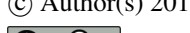



Fig. 6 shows that the large drop of $R_{MSA}$ observed on bulk aerosol in summer (Fig 2, see also Fig. 8 in Sect. 3.2.3.) is consistently revealed by impactor data (open triangles in Fig. 6). It also shows a far more pronounced decrease of $R_{MSA}$ in submicron than micron particles. An example of this strong depletion of MSA relative to nssSO$_4$ in submicron compared to micron particles in summer is seen in Fig. 5c. As a consequence, since the contribution of submicron particles dominates the

5 total mass of MSA, the drop $R_{MSA}$ values in mid-summer at Concordia is mainly due to a drop of $R_{MSA}$ in submicron particles. A drop of $R_{MSA}$ may result from an increase of sulfate and/or a drop of MSA. Impactor data corresponding to the March-November time period (Fig. 7) show that $R_{MSA}$ is very poorly related to the nssSO$_4$ content ($R^2$ of 0.01 and 0.06 for submicron and micron particles, respectively). Conversely, higher is the MSA content higher is $R_{MSA}$ ($R^2$ of 0.55 and 0.47 for submicron and micron particles, respectively). In fact, as seen in Fig. 7, when distinguishing samples having a high and

10 low nssSO$_4$ content (higher and lower than 100 ng m$^{-3}$), a strong relationship is found between MSA and $R_{MSA}$. For instance in submicron particles, $R^2$ equal to 0.86 is calculated for samples containing less than 100 ng m$^{-3}$ of nssSO$_4$ (0.84 for those having more than 100 ng m$^{-3}$ of nssSO$_4$). The same is seen for micron particles ($R^2$ equal to 0.89 and 0.83 for samples containing less and more than 100 ng m$^{-3}$ of nssSO$_4$, respectively).

### 3.2.3. Chemical signature of air mass experienced a summer drop of MSA concentrations

The year-round record of $R_{MSA}$ (HV filter data) was examined at the light of different parameters related to the history of air mass present during sampling at Concordia (Fig. 8). This was done by using 10-day backward trajectory as well as chemical characteristics (i.e. sodium, MSA, nssSO$_4$, and ozone) and air temperature at the site. We restrict the discussion to the last 5 years for which the chemical records are the most continuous (Fig. 8). Sodium is here used to evaluate the importance of marine advection from spring to fall and its inter-annual variability. As discussed by Legrand et al. (2009), ozone at

Concordia exhibits a seasonal cycle characterized by a maximum in July followed by a decrease until October, and the occurrence of a secondary maximum in November-January (Legrand et al., 2009), similarly to what is observed at South Pole (Crawford et al., 2001). Whereas it is expected that such very remote regions experience winter accumulation of O$_3$ transported from other regions followed by photochemical destruction in spring and summer, the occurrence of a secondary maximum in November-January is surprising. That was attributed to a photochemical ozone production induced by the high

NO$_x$ levels generated by the photo-denitrification of the antarctic snowpack (Davis et al., 2001).

The sodium record reported in Fig. 8F does not support the assumption that the drop of $R_{MSA}$ values coincides with a weakening of marine air advections. Indeed whereas there are in general much more sodium (late October/early November) just before the drop of $R_{MSA}$ (see the grey areas in Fig. 8C), the recovery of high $R_{MSA}$ values in fall (February/March) is never accompanied by a recovery of sodium levels. For example, $R_{MSA}$ jumped from < 0.10 mid-February 2014 to 0.40 end-

30 February 2014, whereas the sodium levels remained over the whole period between 1 and 3 ng m$^{-3}$. Consistently with what is observed on impactor (Fig. 7), HV data indicate that the drop of $R_{MSA}$ in mid-summer at Concordia is related to a decrease of MSA rather than an increase of nssSO$_4$ levels (Fig. 8D). The lack of a recovery of marine air advections in fall and the fact that low $R_{MSA}$ values are more related to low MSA rather than high nssSO$_4$ levels do not support the preceding assumption



of a chemistry favoring sulfate during transport in mid-summer as resulting from a weakening of marine advection and a chemistry of DMSO promoting formation of sulfate in the absence of heterogeneous chemistry. Note also that the temperature record (Fig. 8E) does not support the possibility of a drop of MSA caused by evaporative loss from aerosol.

The fact that the drop of $R_{MSA}$ values in mid-summer at Concordia is mainly due to the des-appearance of MSA in the fine aerosol and not to an increase of sulfate, permits to reject the assumptions of (1) a selective deposition of MSA with respect to sulfate or (2) a preferential production of sulfate with respect to MSA during transport between the coast and the inland Antarctic plateau. As seen in Fig. 8, the only significant change that coincides fairly well the drop of $R_{MSA}$ in mid-summer is the occurrence of the secondary maximum of ozone mixing ratio that is attributed to a local photochemical activity driven by $NO_x$ emissions from the snowpack of the Antarctic plateau. To illustrate the timing and the amplitude of the photochemical

ozone production we have reported in Fig. 8A the de-seasonalized ozone record. It is seen that the sudden appearance of low $R_{MSA}$ values that generally occurred beginning of November and ended in February (see the grey areas in Fig 8) coincides with the periods over which the excess of ozone related to the strong photochemical activity took place. The link between low $R_{MSA}$ values and high photochemical activity is also seen in the inter-annual variability, with particularly low $R_{MSA}$ values in Nov-Dec 2011 and 2012 (~0.03) compared to Nov-Dec 2013 and 2014 (0.08) corresponding to larger excess ozone

(10.5 ppbv in Nov-Dec 2011 and 2012 against 8.0 ppbv in Nov-Dec 2013 and 2014). As discussed by Legrand et al. (2016), longer was time spent by the air mass above 3200 m elevation prior to its arrival at Concordia, higher was ozone mixing ratio. That can be seen in Fig. 8B when comparing the fraction of time spent by the air mass above 3200 m asl in Nov-Dec 2011 and 2012 with Nov-Dec 2013 and 2014 (6 days instead of 4-5 days).

### 3.2.4. A destruction of MSA over the Antarctic plateau under mid-summer conditions

The preceding observations of a drop of $R_{MSA}$ driven by a decrease of MSA level in submicron particles around beginning of November and its recovery in February, simultaneous to the high photochemical activity at mid-summer at Concordia, suggests the occurrence of a (photo-)chemical destruction of MSA taking place in submicron particles at that time. A significant in cloud destruction of MSA is suspected to take place in the marine boundary layer (Von Glasow and Crutzen, 2004; Barnes et al., 2006; Hoffmann et al., 2016). Assuming that aerosol particles spend about 3 h per day as cloud droplets

and an aqueous phase OH ($OH_{aq}$) concentration of 6 $10^{-13}$ M, Zhu et al. (2005) calculated a mean lifetime of MSA of 14 days in the marine boundary layer. Clearly, conditions encountered over the Antarctic plateau are very different and it is out of the scope of this paper to identify the involved chemical mechanisms leading to a destruction of MSA in submicron sulfuric acid particles over central Antarctica. We can note however that over the Antarctic plateau, whereas chance for aerosol to experience aqueous phase chemistry in cloud droplets is far lower than in the marine boundary layer, the production of $OH_{aq}$

from the reaction of ozone with $O_2^-$ would be favoured compared to conditions encountered in the marine boundary layer due to far more acidic conditions (Ervens et al., 2003). Note also that the dissolution of $H_2O_2$ that also contributes to the budget of $OH_{aq}$ would be two orders of magnitude higher at Antarctic temperatures than at those encountered in the marine boundary layer.




### 3.3 Biogenic versus non-marine-biogenic source of sulfate inland Antarctica

### 3.3.1. Estimation of non-biogenic sulfate levels from radionuclide data

Apart from marine biogenic emissions, sulfate present over Antarctica can also originate from southern hemisphere continents or the stratospheric reservoir. $^{210}$Pb data permit to derive an estimate of the contribution of sulfate long-range

transported from continents by comparing the $^{210}$Pb concentrations at Concordia (27 µBq m$^{-3}$) after having corrected them from marine $^{222}$Rn exhalation (around 15%, Weller et al. 2014) with those observed at Chacaltaya (407 µBq m$^{-3}$, Feely et al., 1988), a remote site located at 5220 m asl in Bolivia. At this site, a typical sulfate level of 250 ng m$^{-3}$ can be assumed (see Minikin et al. (1998) and references therein). Note that this value is similar to the one reported by Huebert and Lazrus (1980) for the free troposphere (240 ng m$^{-3}$, at 5-6 km elevation over the Pacific ocean). Assuming a sulfate concentration of 250 ng

m$^{-3}$ for the continental free troposphere of the southern hemisphere, and applying a dilution factor of 18 based on $^{210}$Pb data, we calculate a mean sulfate concentration of 14 ng m$^{-3}$. The $^{210}$Pb activities at Concordia (Fig. 2) show an enhancement by a factor of 2.5 from June-September to November-February, leading to an estimated concentration of continental sulfate of 8 and 20 ng m$^{-3}$, respectively. These values may be overestimated since an influence of the city of La Paz (3600 m asl, located only at 25 km away from the Chacaltaya site) on the sulfate concentration remains here possible. Note also that, other

continents such as Australia, certainly contribute to the long-range transport of continental $^{210}$Pb and sulfate towards Antarctica, in particular in the case of East Antarctica (Heimann et al., 1990).

For the first time, $^{10}$Be concentrations are documented in the atmosphere of the high East Antarctic plateau (Fig. 9). The seasonal cycle characterized by a winter minimum and a January-February maximum is similar to what was observed at the coastal site of Neumayer by Elsässer et al. (2011) but with a far stronger winter-summer amplitude (a factor of 10 at

20 Concordia instead of 2-3 at NM). This difference is likely reflecting (1) the particularly strong inversion layer in winter at inland Antarctica, (2) higher summer concentrations at 3200 m asl than at the sea level. From the observation of a mean winter concentration of $^{10}$Be of 0.6 10$^4$ atoms m$^{-3}$ at Concordia (Fig. 9) and considering the $^{10}$Be concentration of 10$^7$ and 0.5 10$^7$ atoms m$^{-3}$ observed between 11 and 19 km elevation at 65°N by Raisbeck et al. (1981) and Field et al. (2006), respectively, we derive a dilution factor in the range of 800-1700 between the lower stratosphere and the atmosphere at

25 Concordia in winter. Lazrus et al. (1979) measured worldwide background (non-volcanic) sulfate mixing ratios from 0.1 ppbm at 11 km elevation to 0.5 ppbm at 19 km elevation. Considering a mean sulfate mixing ratio of 0.3 ppbm for the lower stratosphere, we estimate that stratospheric-tropospheric exchange may account for 0.4 ng m$^{-3}$ of sulfate in winter at Concordia. As shown in Table 3, similarly to what was previously shown for NM (Wagenbach, 1996), an increase of $^7$Be/$^{210}$Pb and $^{10}$Be/$^7$Be ratios from winter to summer is seen at Concordia, suggesting a 2 times stronger downward transport

from the stratosphere in summer than in winter there.

Beryllium data do not account for sulfate downward transported from the lower stratosphere during sedimentation of polar stratospheric clouds that occurs without stratospheric-tropospheric air mass exchange. Measurements of $^{35}$S that offers the possibility to estimate stratospheric input of sulfur were done on aerosol collected at Concordia showing that, as upper





estimates 3.3 ng m$^{-3}$ of sulfate in winter and 16.5 ng m$^{-3}$ in summer/fall come from the lower stratosphere (Hill-Falkenthal et al., 2013).

### 3.3.2. Estimation of non-biogenic sulfate levels from the NssSO$_4$-MSA relationship

The preceding discussions have shown how uncertain still remain estimates of the contribution of non-biogenic source of sulfate over the Antarctic plateau based on radionuclide data, except for the downward transport from the stratosphere as traced back using beryllium ($^{10}$Be and $^{7}$Be).

The initial motivation to conduct simultaneous measurements of MSA and sulfate in Antarctic air or snow was to separate for non-sea-salt sulfate the marine biogenic source from others like volcanic emissions, anthropogenic sources, and terrestrial sources. It was expected that examination of the relationship between nssSO$_4$ and MSA should help here. However, several previous studies pointed out that the quantification of the non-biogenic sulfate sources by examination of the y-intercept of the relationship between nssSO$_4$ and MSA is complicated by the fact that R$_{MSA}$ is seasonally dependent and varies with the MSA concentration range, see for instance Legrand and Pasteur (1998) and Piel et al. (2006). Fig. 10 illustrates how poor is the correlation between the two sulfur species at Concordia ([nssSO$_4$] = 5.5 [MSA] + 14 with R$^2$ = 0.35, Table 4). As recommended by Ayers (2001), we here used a bivariate regression (so called reduced major axis regression, RMA). As clearly shown by Fig. 10a, the scattering of the correlation is largely due to summer samples (R$^2$ = 0.42 for summer samples as shown with red points). Since, as previously discussed, in summer a destruction of MSA takes place at Concordia leading to unusually low R$_{MSA}$ values (even lower than in winter), we have scrutinized the correlation only considering data corresponding to the rest of the year (i.e. considering only black points shown in Fig. 10a). For these samples, that cover the spring to fall time period, a far better correlation is observed ([nssSO$_4$] = 3.4 [MSA] + 7 with R$^2$ = 0.12, Table 4). However, the significant change of R$_{MSA}$ from spring to fall still leads to an overestimation of non-biogenic sulfate when considering the y-intercept of the linear regression line. In the following, we therefore scrutinize separately spring, fall, and winter data (red triangles, black triangles, and blue circles in Fig. 11, respectively).

In winter, both nssSO$_4$ and MSA exhibit low concentrations (Fig. 2). Nevertheless, as suggested by increases of $^{210}$Pb, the isolation of the high Antarctic plateau from the free Antarctic troposphere sometimes breaks down, often resulting from arrival of warm air associated with a marine intrusion. As seen in Fig. 12, under these conditions, the increase of $^{210}$Pb is accompanied by an increase of sulfate, suggesting that the lower troposphere at Concordia was temporary filled with winter free tropospheric air. Interestingly, Fig. 2 shows that the highest mean winter levels of nssSO$_4$ occurred in 2011 (8.0 ± 3.9 ng m$^{-3}$) and 2013 (8.1 ± 5.6 ng m$^{-3}$), when the $^{210}$Pb levels were also the highest of the record (22-24 µBq m$^{-3}$). Conversely, the lowest nssSO$_4$ mean winter level is observed in 2008 when the level of $^{210}$Pb was particularly low in 2008 (11 µBq m$^{-3}$). In the following we examine the origin of sulfate present in the free tropospheric winter atmosphere. As shown in Fig. 11c, the slope of the linear relationship between nssSO$_4$ and MSA ([nssSO$_4$] = 15.2 [MSA] + 0 with R$^2$ = 0.65, Table 4) corresponds to a R$_{MSA}$ value of 0.066 ± 0.004. Such a R$_{MSA}$ value below 0.10 in winter is typically observed in the remote marine boundary layer at low- to mid-southern latitudes in winter (from April to September); 0.077 at 40°S (Cap Grim, Ayers et al.,





1991), 0.037 at 29°S (Norfolk, Saltzman et al., 1986), and 0.026 at 22°S (New Caledonia, Saltzman et al., 1986). The y-intercept of the linear relationship ($0 \pm 1$ ng m$^{-3}$, Table 4) suggests that in winter, when marine biogenic emissions are located far away from the Antarctic continent, the contribution of non-biogenic source to $nssSO_4$ remains at the best limited to around 1 ng m$^{-3}$ (for a total $nssSO_4$ concentration of $6 \pm 4$ ng m$^{-3}$).

In spring, the linear relationship between $nssSO_4$ and MSA ($[nssSO_4] = 5.1$ [MSA] $+ 9$ with $R^2 = 0.89$, Table 4) suggests an increase of $R_{MSA}$ compared to winter (0.20 instead of 0.07 in winter). This increase of $R_{MSA}$ from winter to spring likely reflects the enhanced contribution of DMS emissions from marine area located south of 50°S as shown by the increase of oceanic chlorophyll (Fig. 2). In contrast to winter, the linear relationship between $nssSO_4$ and MSA observed in spring suggests the existence of a non-biogenic sulfate source accounting for a few ng m$^{-3}$. The non-linearity of the relation between

$nssSO_4$ and MSA tends however to overestimate the y-intercept (Fig. 11a), a y-intercept of around 3 ng m$^{-3}$ being obtained when samples having less than 5 ng m$^{-3}$ of MSA are considered.

   In fall, the linear relationship between $nssSO_4$ and MSA becomes even less linear than in spring, rendering rather inaccurate the use of the y-intercept in evaluating the non-biogenic sulfate contribution (Fig. 11b). However, Fig. 11b suggests that, if significant, the non-biogenic sulfate concentration remains well below 5 ng m$^{-3}$ at that time. The slope of the relationship

between $nssSO_4$ and MSA in fall ($2.8 \pm 0.2$, Table 4) indicates a further increase of $R_{MSA}$ compared to spring (0.36 instead of 0.20 in spring). Comparing November and March, Fig. 2 indicates that a decrease of oceanic chlorophyll concentrations have started between 50 and 60°S whereas those at latitudes higher than 60°S are maintained or slightly higher in March than in November. That can explain the higher $R_{MSA}$ values in March compared to November. Note that such a higher $R_{MSA}$ in March compared to November is also observed at coastal sites (Fig. 3).

To minimize the uncertainties related to the variability of $R_{MSA}$ between the different samples, we also examined the relationship between non-sea-salt sulfate and MSA concentrations on the 12 stages of impactor run. This approach reduces the uncertainty linked to variability of $R_{MSA}$ value over time. A few impactor run show a y-intercept that was significantly different from zero, reaching 1 ng m$^{-3}$ in winter and 1 to 4 ng m$^{-3}$ at other seasons.

   It can therefore be concluded that, whatever the season, marine biogenic emissions of DMS dominate the atmospheric

budget of sulfate over inland Antarctica. If it exists, the contribution of non-biogenic sulfate source remains limited to 1 ng m$^{-3}$ in winter, and possibly reaches a few ng m$^{-3}$ particular from spring to fall. Previous discussions on radionuclide data gained at Concordia suggest the long-range transported sulfate from continent in spring/summer and downward transport from the lower stratosphere particularly in summer/fall as non-biogenic source of sulfate.

### 4. Coastal Antarctica in summer

As seen in Table 5, in summer there is a systematic difference in the sulfur aerosol composition at sites facing the Atlantic sector compared to those facing the Indian sector. Whereas a mean $R_{MSA}$ values close to 0.3 is observed at all sites in March, a summer $R_{MSA}$ maximum of 0.4 occurs in January at NM and Halley (Atlantic sector) when the $R_{MSA}$ value remains close to



0.2 at DDU and Mawson (Indian sector). The corresponding relatively weak abundance of MSA compared to nssSO$_4$ in January at DDU/Mawson compared to NM/Ha may be related to a destruction of MSA that acts more efficiently at the Atlantic than Indian margin regions. Indeed the level of various oxidants was found to be unusually high at DDU compared to the situation at Ha. For instance, Kukui et al. (2012) reported a mean OH concentration of 2.1 10$^6$ radical cm$^{-3}$ at DDU

against 0.39 10$^6$ radical cm$^{-3}$ at Halley (Bloss et al., 2010). Note that OH concentrations at DDU are still in the range of those observed at Concordia (3.1 10$^6$ radical cm$^{-3}$, Kukui et al., 2014). Concerning ozone, Legrand et al. (2016) compared records from NM and Halley with those of DDU and Syowa, a site also facing the Indian sector. They found that in December, more frequently high ozone values (17% above 25 ppbv, 33% above 22 ppbv) are observed at DDU compared to NM (1% above 25 ppbv, 5% above 22 ppbv) and HA (< 1% above 25 ppbv, 1% above 22 ppbv). For the case of Syowa, an intermediate

situation is observed with 2.5 % of values above 25 ppbv, and 12% of values above 22 ppbv. These differences were attributed to the fact that ozone-rich air masses present in summer over the inland Antarctic plateau are more efficiently transported to DDU and Mawson than to NM and Halley. Since the near-surface airflow between the Antarctic plateau and the coastal regions is largely controlled by the topography of the underlying ice sheets and the vicinity of low-pressure systems on the coast of the Antarctic continent, the transport of air mass from inland Antarctica to margin regions is far more

important at DDU and Mawson than at NM and Halley (Parish and Bromwich, 2007). These differences in the oxidative property of the atmosphere may lead to larger destruction of MSA and therefore to a decrease of the R$_{MSA}$ ratio at DDU and Mawson compared to NM and Ha, as seen in Table 4.

## 5. Implications for the R$_{MSA}$ ratio in Antarctic ice

In the present-day (2006-2015) aerosol at Concordia, we observe an annual mean level of MSA and nssSO$_4$ of 4 and 37 ng

m$^{-3}$, respectively, with a grand average R$_{MSA}$ value of 0.11. Assuming a similar air-snow relationship for MSA and nssSO$_4$ aerosol and referring to the mean level of sulfate observed in the Holocene ice (0-10 kr BP) at Dome C (100 ppb, Wolff et al., 2006), we would expect a corresponding level of MSA in ice of 12 ppb. The lower MSA value observed in the ice deposited during the Holocene at Dome C (from 1 to 5 ppb, Saigne and Legrand, 1987) indicates that, in addition to the previously discussed destruction of MSA in the atmosphere of central Antarctica, post-depositional effects also contribute to the low

R$_{MSA}$ value seen in ice at that site.

Though year-round MSA data are not available at the South Pole, the drop of R$_{MSA}$ values seen at Concordia during mid-summer is also observed at the South Pole (see Table 2) supporting the relevance of an annual R$_{MSA}$ of 0.11 for the atmosphere at the scale of the whole Antarctic plateau. If confirmed, a R$_{MSA}$ value of 0.11 in the atmosphere at the South Pole is consistent with R$_{MSA}$ observed in south polar snow layers, 0.13 over the last century (Legrand and Feniet-Saigne,

1991) and from 0.08 to 0.15 over the last millennium (Feniet, 1984). That also suggested that, if they occur, post-depositional effects remains rather limited at that site.

At Vostok, whereas R$_{MSA}$ values ranging around 0.15 were observed in the upper 2 m of snow (Wagnon et al., 1999), an




averaged value of 0.05 is found in ice deposited over the last 10 kyr BP (Legrand et al., 1991). Therefore, in contrast to the case of the South Pole where atmospheric process can alone explained the relatively low $R_{MSA}$ values, at sites characterized by lower snow accumulation rates like Vostok and Concordia (2 and 3 g cm$^{-2}$ yr$^{-1}$, respectively instead of 8 g cm$^{-2}$ yr$^{-1}$ at the South Pole), there are also post-depositional destruction or release of MSA within the snowpack. Further measurements
including gas phase MSA would be needed at Concordia to conclude on the causes of post-deposition decrease of MSA in snow (re-emission into the gas phase or in situ chemical destruction).

## 6. Conclusions

Load and composition of sulfur-derived aerosol (methanesulfonate and non-sea-salt sulfate) at inland East Antarctica are documented from multiple year-round records of bulk aerosol samplings, and for the first time in central Antarctica, the size-
segregated composition of aerosol (0.03-20 micron diameter). A striking difference in the seasonality of sulfur aerosol composition, characterized by a MSA to nssSO$_4$ ratio reaching a minimum in December-January over the Antarctic plateau (0.05) and a maximum at the coast (up to 0.40 at sites facing the Atlantic oceanic sector), is clearly established. We find that the low value of $R_{MSA}$ in mid-summer at Concordia is due to a drop of MSA concentrations that occurs in the small particles (0.3 μm diameter) of sulfuric acid aerosol. The drop of MSA coincides with periods of high photochemical activity as
indicated by the presence of ozone locally photo-chemically produced, strongly suggesting the occurrence of an efficient chemical destruction of MSA over the Antarctic plateau in mid-summer. The examination of the relationship between MSA and nssSO$_4$ levels indicate a non-biogenic sulfate level that does not exceed 1 ng m$^{-3}$ in fall and winter and remains below 5 ng m$^{-3}$ in spring. Thanks to atmospheric $^{10}$Be, $^{7}$Be, and $^{210}$Pb data gained at Concordia, this weak level of non-biogenic sulfate over the Antarctic plateau is discussed with respect to the contribution of the stratosphere and of the long-range
transport of sulfate from continents. The observed increases of $R_{MSA}$ values from winter to spring and spring to fall reflect change over the course of the year of marine source regions contributing to the sulfur load inland Antarctica, with marine emissions mainly located at temperate latitudes in winter and a progressive recovery of high-latitude DMS emissions in spring-summer-fall. The findings demonstrate that the relatively low $R_{MSA}$ observed in the ice deposited over the plateau compared to that at coastal Antarctica reflects at the first degree the atmospheric behaviour of sulfur derived aerosol,
although, at least at low snow accumulation rate sites, a loss of MSA from the snowpack due to either chemical destruction or reemission occurs there.

## Data availability

Data on the chemical composition of aerosol (bulk and size-segregated composition) at Concordia can be made available for scientific purposes upon request to the authors (contact: michel.legrand@univ-grenoble-alpes.fr or
Suzanne.Preunkert@univ-grenoble-alpes.fr). Original $^{10}$Be and $^{9}$Be can be obtained by contacting HZDR



(s.merchel@hzdr.de).

**Acknowledgements**

National financial support and field logistic supplies for the summer campaign were provided by the Institut Polaire Français-Paul Emile Victor (IPEV) through program n°414 and 903 and the Agence Nationale de la Recherche through contract ANR-14- CE01-0001-01 (ASUMA). This work was initiated in the framework of the French environmental observation service CESOA (Etude du cycle atmosphérique du Soufre en relation avec le climat aux moyennes et hautes latitudes Sud, http://www-lgge.obs.ujf-grenoble.fr/CESOA/spip.php?rubrique2) with the financial support of INSU (CNRS). We thank Bruno Jourdain from LGGE for supervising the sampling material in the field and for sample analysis. Thanks also to Eric Wolff from Cambridge for useful discussions. Parts of this research were carried out at the Ion Beam Centre (IBC) at the Helmholtz-Zentrum Dresden-Rossendorf e. V., a member of the Helmholtz Association. We would like to thank the DREAMS operator team, R. Ziegenrücker and S. Pavetich for their assistance with AMS-measurements and S. Uhlig for help with $^{10}$Be sample preparation.

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



**Tables**

**Table 1:** MSA, nssSO$_4$, and MSA/nssSO$_4$ ratios (R$_{MSA}$) in winter (June-September), November, and January at Concordia
(DC, 2006-2015) and the coastal site of Neumayer (NM, 1983-1995) and Dumont d'Urville (DDU, 1991-1996).

| Sites/Periods | MSA (ng m$^{-3}$) | NssSO$_4$ (ng m$^{-3}$) | R$_{MSA}$ | References |
|---|---|---|---|---|
| DC (Jun-Sep) | 0.6 ± 0.4 | 6.4 ± 2.2 | 0.08 ± 0.02 | This work |
| NM (Jun-Sep) | 3.3 ± 1.9 | 40 ± 11 | 0.08 ± 0.02 | Minikin et al. (1998) Legrand and Pasteur (1998) |
| DDU (Jun-Sep) | 2.4 ± 1.2 | 27 ± 8 | 0.09 ± 0.02 | Minikin et al. (1998) Jourdain and Legrand (2002) |
| DC (Nov) | 5.6 ± 1.9 | 64 ± 22 | 0.09 ± 0.02 | This work |
| NM (Nov) | 19.6 ± 6.1 | 152 ± 32 | 0.13 ± 0.03 | Minikin et al. (1998) Legrand and Pasteur (1998) |
| DDU (Nov) | 17 ± 2.5 | 151 ± 33 | 0.11 ± 0.04[*] | Minikin et al. (1998) Jourdain and Legrand (2002) |
| DC (Jan) | 4.7 ± 2.4 | 84 ± 25 | 0.05 ± 0.02 | This work |
| NM (Jan) | 154 ± 77 | 380 ± 130 | 0.41 ± 0.13 | Minikin et al. (1998) Legrand and Pasteur (1998) |
| DDU (Jan) | 60 ± 23 | 280 ± 79 | 0.21 ± 0.05[*] | Minikin et al. (1998) Jourdain and Legrand (2002) |

[*]The values are slightly higher than those reported by Legrand and Pasteur (1998) (0.08 ± 0.02 in November, and 0.16 ± 0.01
in January) since, following Jourdain and Legrand (2002), they were calculated after having subtracted the contribution of
ornithogenic soils to the sodium and sulfate levels.

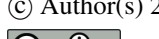



**Table 2:** MSA/nssSO$_4$ ratios (R$_{MSA}$) observed at inland Antarctic sites in mid-summer.

| Site | R$_{MSA}$ | Month/year | References |
|---|---|---|---|
| | 0.063 | Nov - Dec 2003 | Arimoto et al. (2008) |
| South Pole | 0.08 | Nov 2000/Jan 2001 | Arimoto et al. (2004) |
| | 0.059 | Dec 1998/Jan 1999 | Arimoto et al. (2001) |
| | 0.094 | Jan 2000 | |
| | 0.052 | Dec 2000/Jan 2001 | Piel et al. (2006) |
| | 0.15 | Dec 2001/Jan 2002 | |
| Concordia | 0.028 | Dec 2000/Jan 2001 | Udisti et al. ( 2004) |
| (75°S 123°E) | ~0.04 | Dec 2006 | Preunkert et al. (2008) |
| | ~0.40 | Mar 2006 | |
| | 0.05 ± 0.03 | Jan 2006, 2008-2015 | This work |
| | 0.25 ± 0.09 | March 2006, 2008-2015 | |
| EDML | 0.15 ± 0.05 | Jan/Feb 2000-2002 | Piel et al. (2006) |
| (75°S 0°E) | 0.33 | March 2003-2005 | Weller and Wagenbach (2007) |





**Table 3:** Monthly data of atmospheric $^{10}$Be and $^{7}$Be at Concordia along with tracers of arrival of stratospheric aerosols ($^{10}$Be/$^{7}$Be as atom ratio and $^{7}$Be/$^{210}$Pb as activity ratio).

| Months | $^{10}$Be (atoms m$^{-3}$) | $^{7}$Be (atoms m$^{-3}$) | $^{10}$Be/$^{7}$Be (atom) | $^{7}$Be/$^{210}$Pb (activity) |
|--------|------|------|------|------|
| Jan | 6.2 | 3.6 | 2.0 | 122 |
| Feb | 5.2 | na | na | na |
| Mar | 6.0 | na | na | na |
| Apr | 2.3 | na | na | na |
| May | 1.3 | 1.5 | 0.9 | 96 |
| Jun | 0.7 | na | na | na |
| Jul | 0.6 | na | na | na |
| Aug | 0.5 | 0.8 | 0.6 | 82 |
| Sep | 0.8 | 1.4 | 0.6 | 78 |
| Oct | 2.1 | 2.4 | 1.1 | 88 |
| Nov | 3.0 | 2.8 | 1.2 | 85 |
| Dec | 4.8 | 3.8 | 1.5 | 140 |

5    na Not available





**Table 4:** Slope and y-intercept (± standard error estimate) of the linear regression between nssSO$_4$ and MSA levels as a function of season at Concordia.

| Data set | Slope | y-intercept | $R^2$ |
|---|---|---|---|
| All data | 5.5 ± 0.25 | 14 ± 2 | 0.35 |
| All data excluding summer | 3.4 ± 0.1 | 7 ± 1 | 0.79 |
| Summer | 10.0 ± 0.8 | 36 ± 7 | 0.42 |
| Winter | 15.2 ± 1.0 | 0.0 ± 1 | 0.65 |
| Spring | 5.1 ± 0.3 | 9 ± 4 | 0.89 |
| Fall | 2.8 ± 0.2 | 12 ± 5 | 0.75 |





**Table 5:** MSA/nssSO$_4$ ratios (R$_{MSA}$) observed at coastal Antarctic sites in January and March, Neumayer and Halley facing the Atlantic ocean, Dumont d'Urville and Mawson the Indian ocean.

| Site | R$_{MSA}$ | Month/year | References |
|---|---|---|---|
| Neumayer | 0.41 ± 0.13 | Jan 1984-1995 | Legrand and Pasteur (1998) |
| (70°S 8°W) | 0.27 ± 0.10 | Mar 1983-1994 | |
| Halley | 0.43 ± 0.03 | Jan 1992-1993 | Legrand and Pasteur (1998) |
| (75°S 26°W) | 0.35 ± 0.12 | Mar 1991-1992 | |
| Mawson | 0.23 ± 0.05 | Jan 1988-1991 | Legrand and Pasteur (1998), |
| (67°S 62°E) | 0.31 ± 0.07 | Mar 1988-1991 | Savoie et al. (1992) |
| Dumont d'Urville | 0.21 ± 0.05[*] | Jan 1991-1996 | Jourdain and Legrand (2002) |
| (66° S 140°E) | 0.33 ± 0.06[*] | Mar 1991-1996 | |

[*]The values are slightly higher than those reported by Legrand and Pasteur (1998) (0.16 ± 0.01 in January and 0.25 ± 0.04 in March) since, following Jourdain and Legrand (2002), they were calculated after having subtracted the contribution of ornithogenic soils to the sodium and sulfate levels.





**Figures**

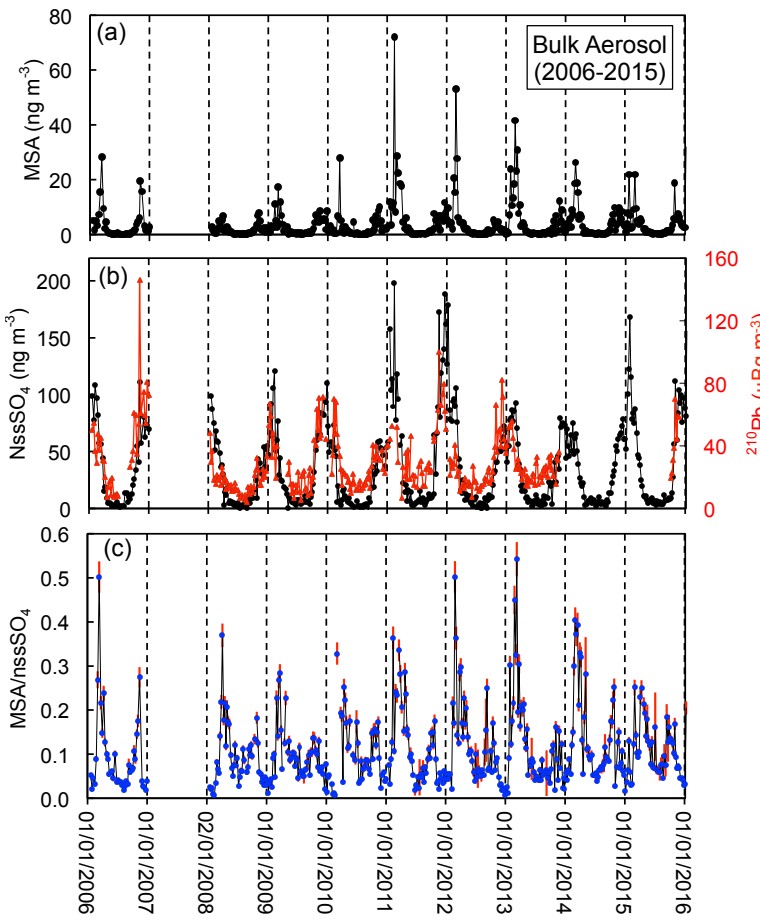

**Figure 1.** Weekly bulk aerosol concentrations of MSA (a), nssSO$_4$ together with $^{210}$Pb activities (b) and the mass MSA/nssSO$_4$ ratio (c). Vertical bars refer to uncertainty in calculating the MSA/nssSO$_4$ ratio (equations 3). Nine values of the MSA/nssSO$_4$ ratio were off scale: August 2008 (1.1 ± 0.8), October 2008 (0.8 ± 0.1), May 2009 (0.7 ± 0.3), March 2010 (1.3 ± 0.1, 4.3 ± 0.4, and 1.7 ± 0.1), June 2010 (2.4 ± 1.3), September 2010 (-1.2 ± 2.2), and July 2012 (1.0 ± 2.2).





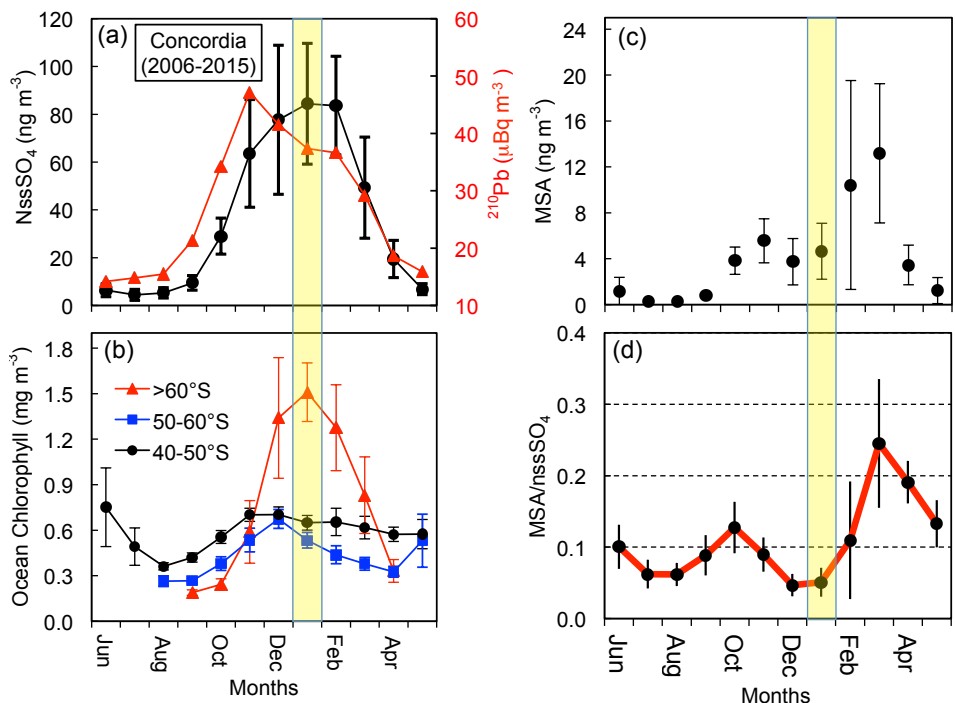

5 **Figure 2.** Monthly mean concentration of nssSO$_4$ along with $^{210}$Pb level (a), MSA (c), and MSA/nssSO$_4$ ratio (d) in bulk aerosol collected at Concordia from January 2006 to January 2016. (b) Monthly mean chlorophyll concentration in the Southern Ocean (2002-2011) (MODIS-Aqua satellite data as reprocessed by Johnson (2013), Johnson et al. (2013)). Vertical bars denote year-to-year variability.





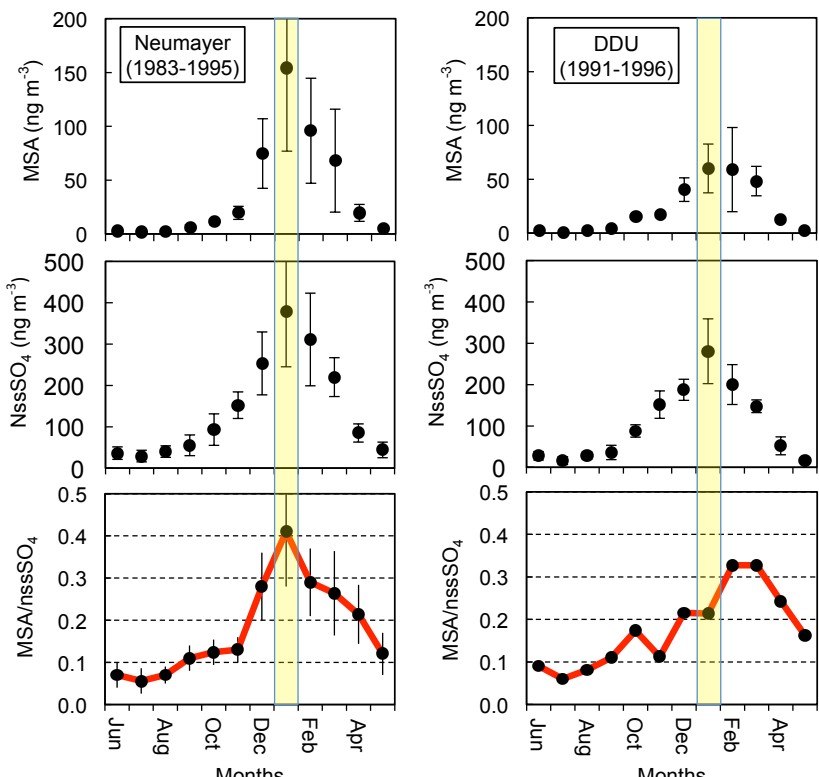

5   **Figure 3.** Monthly mean values of MSA, nssSO$_4$, and MSA/nssSO$_4$ ratio, in bulk aerosol collected at Neumayer (1983-1995) (left) and DDU (1991-1996) (right). Adapted from Minikin et al. (1998) and Legrand and Pasteur (1998). Vertical bars denote year-to-year variability.





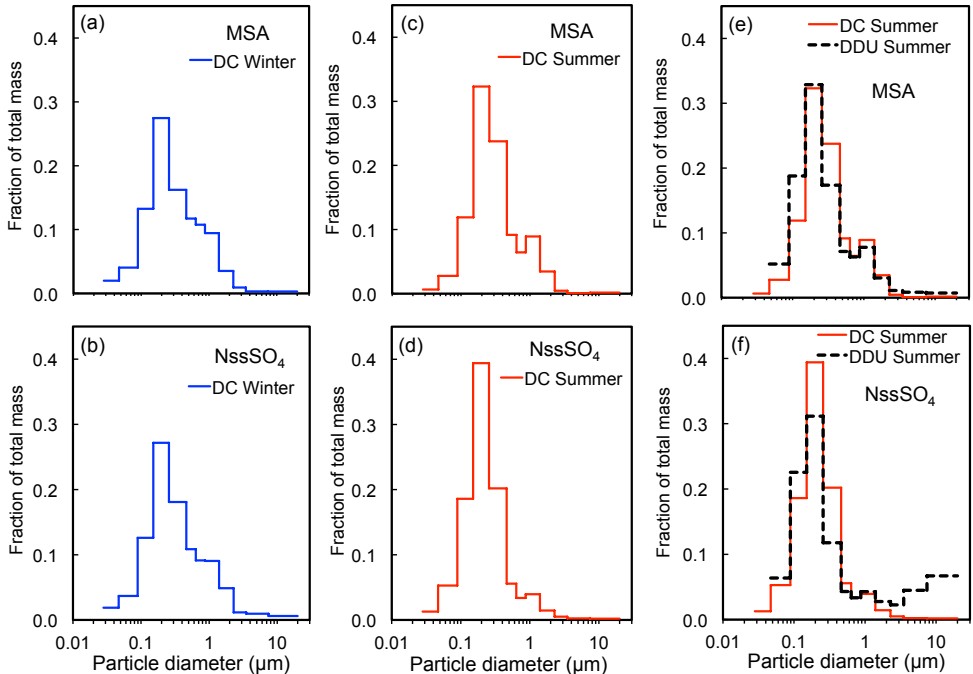

**Figure 4.** Mean size-segregated mass composition of sulfur aerosol (MSA and non-sea-salt sulfate) at Concordia in winter (a and b) and summer (c and d), at Concordia and DDU in summer (e and f). The presence of very large nssSO$_4$ particles at DDU in summer (dashed black line in panel f) is due to sulfate from ornithogenic soils present at the site (Jourdain and Legrand, 2002). Note that at DDU the impactor was run using only 11 stages missing the smallest particles.




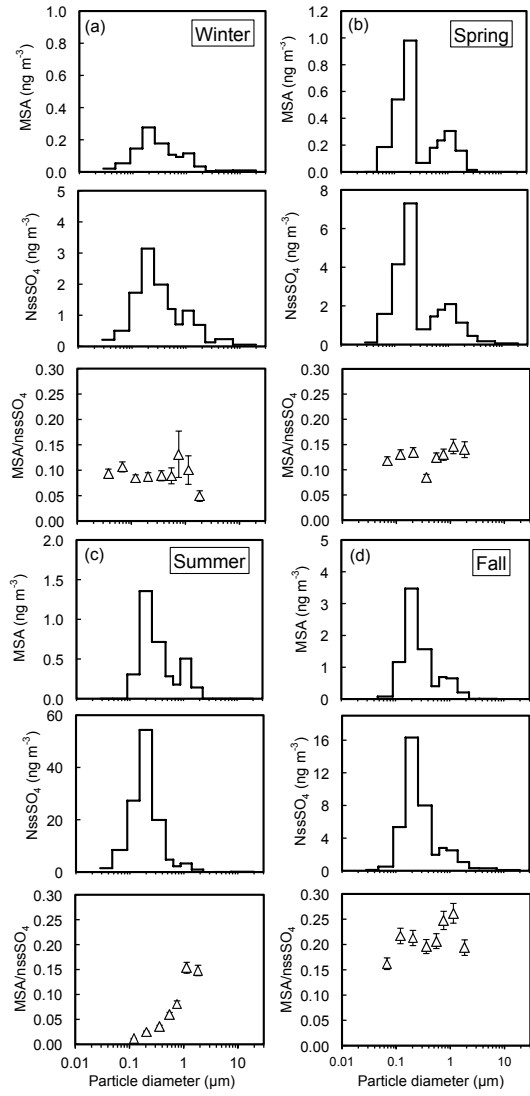

**Figure 5.** Size-segregated composition of sulfur aerosol (MSA, non-sea-salt sulfate, and MSA/nssSO₄) at Concordia in winter (a, from 14 to 28 August 2009), spring (b, 9 to 22 October 2010), summer (c, 9 to 22 January 2010), and fall (d, 28 March to 11 April 2011).





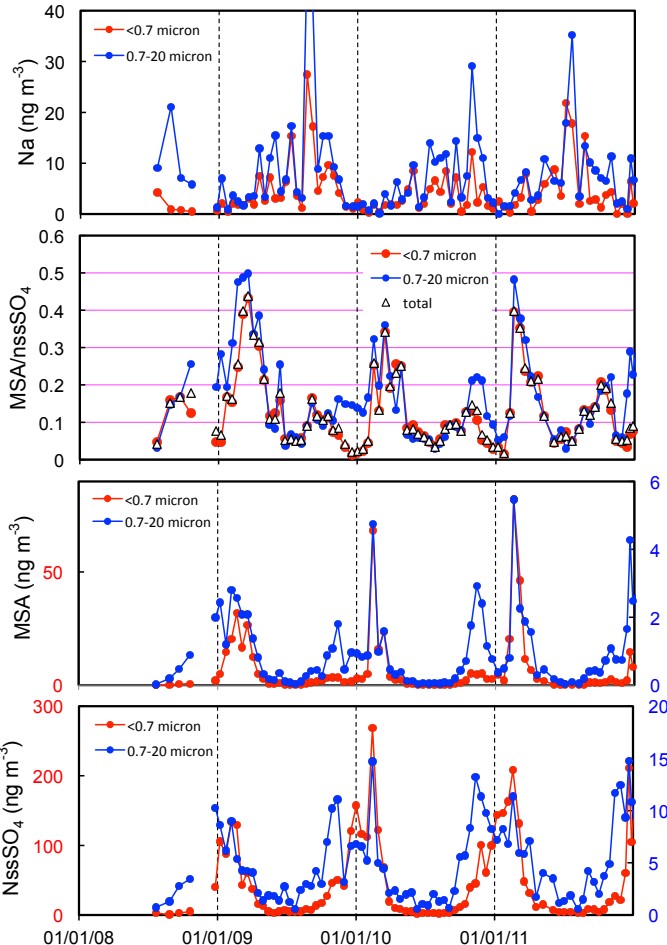

**Figure 6.** Year-round records of the chemical composition of aerosol collected from 2008 to 2011 at Concordia on the 12-stage impactor, distinguishing between small (the last 6 stages, i.e. 0.08-0.7 μm diameter) and large (the first 6 stages, i.e. 0.7-20 μm diameter) particles. From top to bottom: sodium, MSA/nssSO$_4$ ratio, MSA, and nssSO$_4$. For the MSA/nssSO$_4$ ratio, we also report values corresponding to the total mass of collected aerosol (open triangles). Note the different scales used for small (left scales) and large (right scales) particles for MSA and nssSO$_4$.





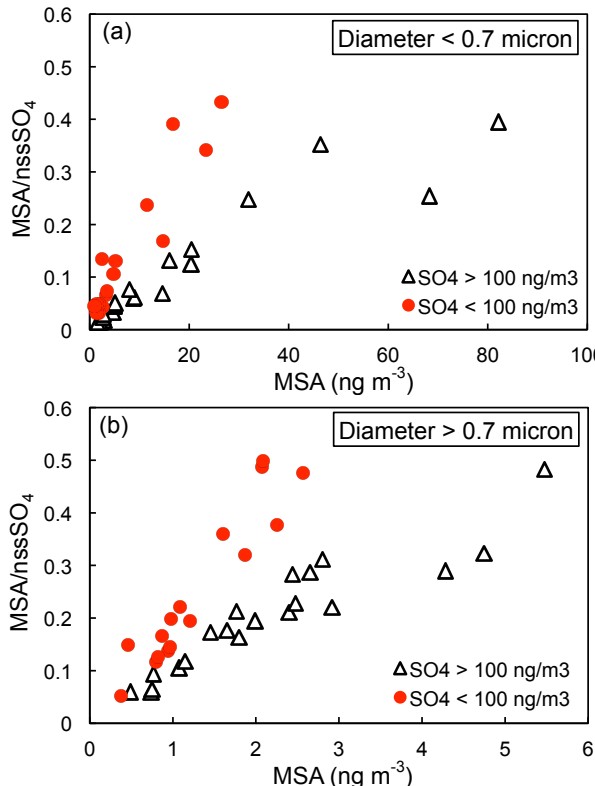

**Figure 7.** Relationship between the MSA/nssSO$_4$ ratio and the MSA level in small (a) and large (b) particles collected on the 12-stage impactor in summer (from November to March). In both cases, we distinguish samples containing more (open triangles) or less (red points) than 100 ng m$^{-3}$ of nssSO$_4$.



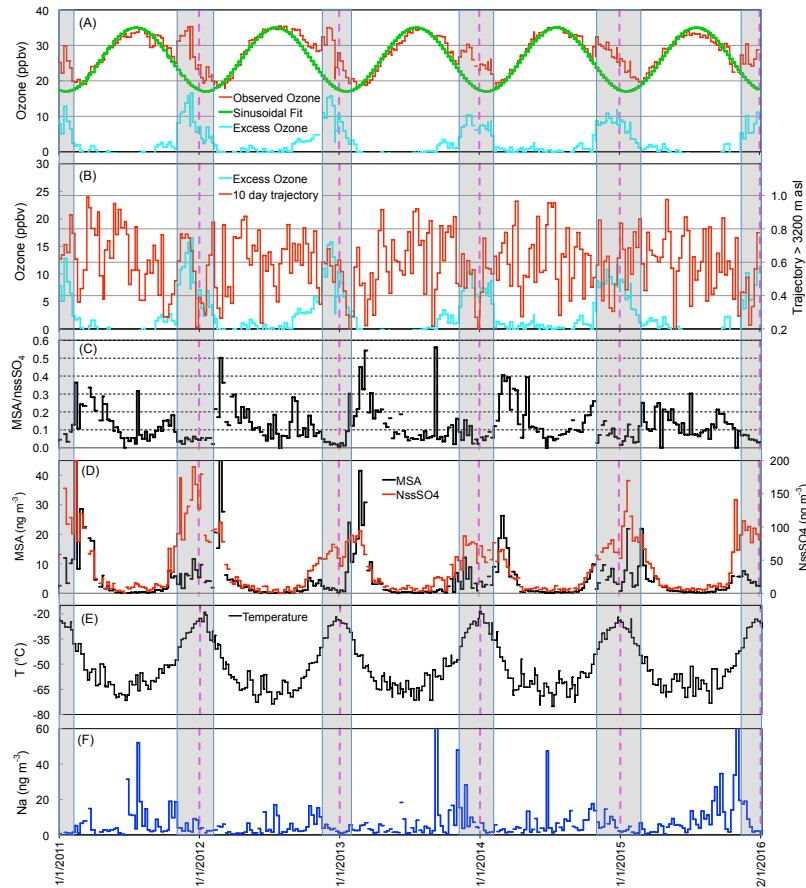

**Figure 8.** (A): Ozone mixing ratio corresponding to HV aerosol sampling time (red line), a sinusoidal fit of the ozone seasonal cycle (green line) and the amount of ozone present in excess (turquoise line) in summer are reported (see Sect. 3.2.3). (B): Excess ozone (turquoise line) together with the 10-day backward trajectory (arrival at 0 m asl) at Concordia (see details in Legrand et al., this issue). The red curve in (B) is the fraction of time spent above 3200 m asl by the air masses arriving at Concordia. (C) and (D): MSA/nssSO$_4$ mass ratio and MSA along with nssSO$_4$ observed on HV samples, respectively. (E): Air temperature at Concordia. (F) Sodium on HV samples. The grey area denotes the fast decreases of R$_{MSA}$ that fairly well coincide with the local O$_3$ photochemical production.





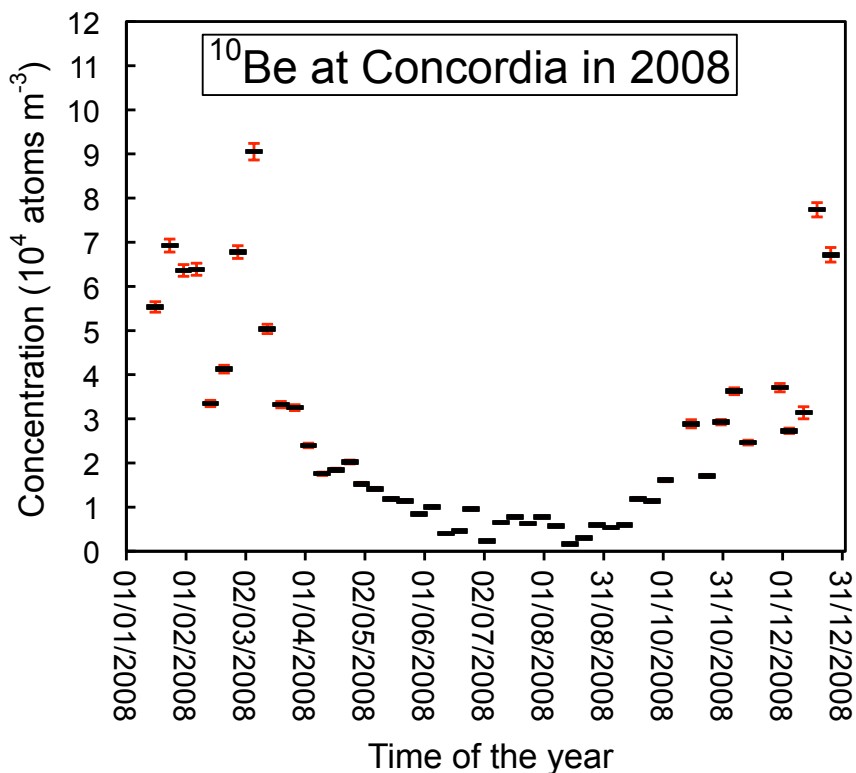

5 **Figure 9.** Annual cycle of [10]Be concentrations in 2008 at Concordia. Vertical bars (in red) refer to AMS uncertainties (see Sect. 2).





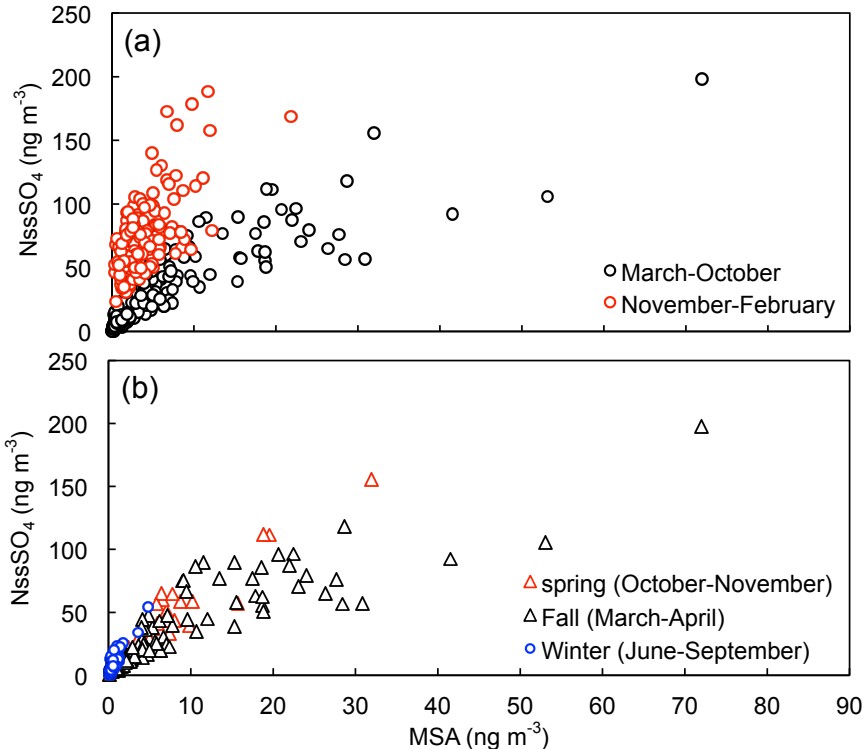

**Figure 10.** Correlation of nssSO$_4$ with MSA concentrations observed at Concordia on HV bulk aerosol samples. (a): All data
5   (red circles highlight mid-summer samples), (b): Spring to fall samples.




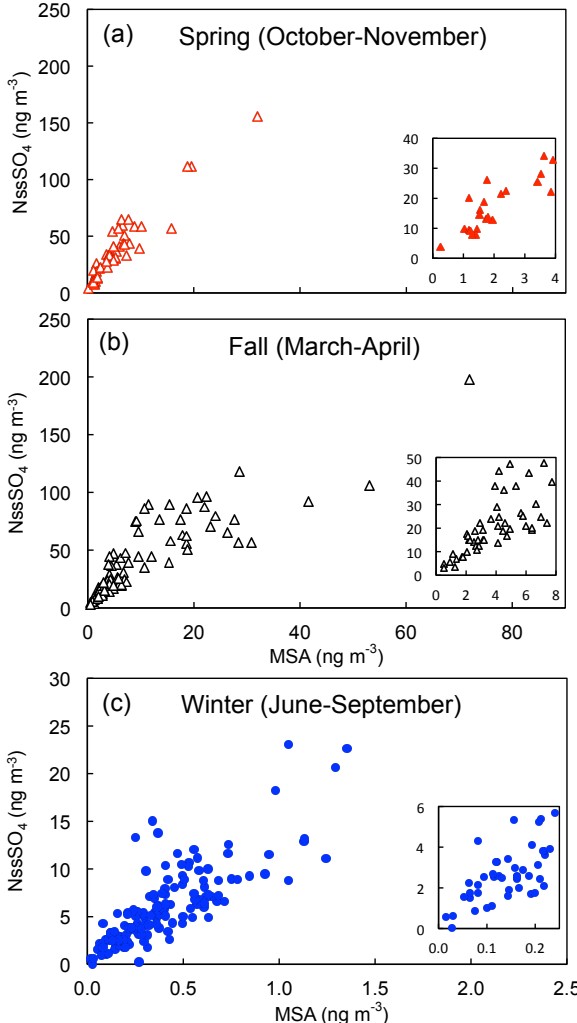

**Figure 11.** Correlation of nssSO$_4$ with MSA concentrations observed at Concordia on HV bulk aerosol samples collected in spring (a), fall (b) and winter (c). The panelled-in figures highlight the correlation at MSA concentrations close to zero.



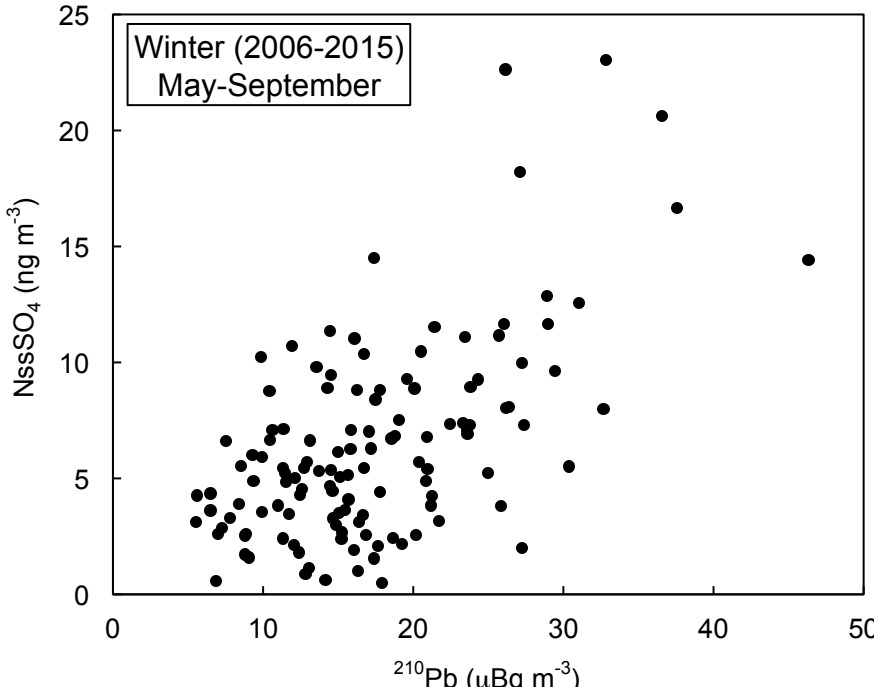

**Figure 12.** NssSO$_4$ versus $^{210}$Pb concentrations on HV filters collected in winter (May-September) at Concordia between 2006 and 2015.