# Peer review of "Year-round record of bulk and size-segregated aerosol composition in central Antarctica (Concordia site) Part 2: Biogenic sulfur (sulfate and methanesulfonate) aerosol"

_Atmospheric Chemistry and Physics, 2017_

## Referee Comment (RC1) · Anonymous Referee #1 · 31 May 2017

This manuscript discusses the sources and processes affecting sulfate and MSA concentrations, and the MSA to sulfate concentration ratio, in the continental Antarctica. This is a very important topic, since the MSA to sulfate ratio has been widely used in estimating biogenic vs anthropogenic sulfur source sources as well as in interpreting ice core data. The measurement data used in this paper appears to be of good quality and the analysis itself is, in most part, scientifically sound. I have a few minor issues to be considered before accepting this paper for publication.

Main comments:

End of section 3.2.2 and Figure 7: separating the data into those with sulfate concentration lower and larger than 100 ng/m3 seems very artificial. What was the bases for this specific border? Do the authors have any idea why the two subsets showed much higher correlations than all the data together?

The discussion in section 3.2.4 is highly speculative. Furthermore, the authors only mention the destruction of MSA in cloud water in this section, even though multi-phase reactions have also reported to be an important source of MSA as also mentioned in section 3.2.1. The subsection 3.2.4 should be partially rewritten.

Page 10, lines 11-14: I have some difficulty in following the logic here. Please clarify, especially what you mean by overestimation here and what is the actual reason for it.

Page 12: lines 5-13: The linear fits given in Table 4 should also be presented in Figure 11. Besides correlations, how significant were the derived relations? A statement such as "the linear relationship . . . even less linear. . ." is meaningless/incorrect in a statistical sense and should be modified.

Minor issues:

Page 3, line 12: I do not think that this is an acceptable way of citing other ACP papers (Legrand et al. this issue).

Page 5, line 15: weakness in abundance? Strange wording.

section 3.2.2: should it be supermicron particles rather than micron particles?

Page 8, line 8: . . .the higher. . .the higher. Please check out the grammar

Page 9, line 4: des-appearance. Please check out the wording.

---

## Referee Comment (RC2) · Anonymous Referee #2 · 10 Oct 2017

GENERAL

The paper presents measurements, results and analyses of sulfur aerosols at the Concordia station. The work is carefully done, it is a valuable paper for the interpretation of Antarctic aerosols and ice cores. As a highlight I would mention the interesting result of the interpretation of the MSA/nssSO4 and the photochemical destruction of MSA in summer. I can recommed publishing the paper in ACP, I only have minor revision suggestions.

[Figure]

DETAILED COMMENTS

The time series is fairly long – are there any statistically significant trends? Yes or no, it would be potentially important.

In the methods section: - sulfate might also come from the stations generator – could it? - was there any sector control?

P4L8-11 "sulfate depletion relative to sodium with respect to the seawater composition .." there is the reference to the full paper but you could add a sentence or two as an explanation of the depletion here, too.

Section 3.2.2 I am missing some comparison of HV and impactor data. I guess it has been done. A scatter plot with explanations would be nice.

In Fig 4: there are the average size distributions of the respective seasons. How about showing there some range? Also Becagli et al. (Atmos. Environ., 52, 98–108, 2012) showed size distributions measured at Dome C – make some comment on the main differences.

P8,L6-8 "Impactor data corresponding to the March-November time period (Fig. 7) show that RMSA is very poorly related to the nssSO4 content (R2 of 0.01 and 0.06 for submicron and micron particles, respectively)". Fig 7 shows R vs MSA, not R vs nssSO4. I suggest adding subfigures where this is shown.

P10L9-10 "Assuming a sulfate concentration of 250 ng m-3 for the continental free troposphere of the southern hemisphere, and applying a dilution factor of 18 based on 210Pb data" Please explain how the dilution factor of 18 was obtained. Any uncertainty estimate for it?

P10L26-27 "Considering a mean sulfate mixing ratio of 0.3 ppbm for the lower stratosphere, we estimate that stratospheric-tropospheric exchange may account for 0.4 ng m-3 of sulfate" Is 0.3 ppb = 0.4 ng/m3?

Table 2 shows R in midsummer. March is not really midsummer any more.

Figures with scatterplots: why don't you show the regressions there?

---

## Author Comment (AC1) · 20 Oct 2017

This manuscript discusses the sources and processes affecting sulfate and MSA concentrations, and the MSA to sulfate concentration ratio, in the continental Antarctica. This is a very important topic, since the MSA to sulfate ratio has been widely used in estimating biogenic vs anthropogenic sulfur source sources as well as in interpreting ice core data. The measurement data used in this paper appears to be of good quality and the analysis itself is, in most part, scientifically sound. I have a few minor issues to be considered before accepting this paper for publication.

***We would like first to thank the reviewer for its helpful comments (see our detailed answers below). We identify 10 bulk sulfate and MSA erroneous values (in January/March 2009 and 2010). They were corrected and all figures and Tables accordingly. These erroneous values however did not change the overall results and conclusions.***

Main comments:

End of section 3.2.2 and Figure 7: separating the data into those with sulfate concentration lower and larger than 100 ng/m3 seems very artificial. What was the bases for this specific border? Do the authors have any idea why the two subsets showed much higher correlations than all the data together?

**We updated Figure 7 that now also shows the plots versus sulfate.**
**There is no strict border but as now explained in the text separating data with high and low sulphate permit to check the drop of RMSA from November to December and from February/March to January. "Note that most of samples containing less than 100 ng m$^{-3}$ of nssSO$_4$ correspond to the November-December period, whereas those containing more than 100 ng m$^{-3}$ of nssSO$_4$ to the January-March period. In this way, the drop of $R_{MSA}$ from November to December and from February/March to January is examined separately. In both cases, the $R_{MSA}$ drop is mainly due to a decrease of MSA. »**

The discussion in section 3.2.4 is highly speculative. Furthermore, the authors only mention the destruction of MSA in cloud water in this section, even though multi-phase reactions have also reported to be an important source of MSA as also mentioned in section 3.2.1. The subsection 3.2.4 should be partially rewritten.

**Well, we stated at the end of Section 3.2.3 that "The fact that the drop of $R_{MSA}$ values in mid-summer at Concordia is mainly due to the disappearance of MSA in the fine aerosol and not to an increase of sulfate, permits to reject the assumptions of (1) a selective deposition of MSA with respect to sulfate or (2) a preferential production of sulfate with respect to MSA during transport between the coast and the inland Antarctic plateau. As seen in Fig. 8, the only significant change that coincides fairly well the drop of $R_{MSA}$ in mid-summer is the occurrence of the secondary maximum of ozone mixing ratio that is attributed to a local photochemical activity driven by NO$_x$ emissions from the snowpack of the Antarctic plateau." It is therefore legitimate to question the occurrence of a chemical destruction of MSA as we did in this section 3.2.2.**

**But you are right we forget to mention that, in contrast to the fast heterogeneous production of MSA, under conditions encountered in the marine boundary layer its destruction is slow. But once formed, MSA can be subsequently destroyed over the Antarctic plateau given the high levels of oxidants there and the few days during which the air mass travels over this region. We updated the discussion here as follows: "The preceding observations of a drop of $R_{MSA}$ driven by a decrease of MSA level in submicron particles around beginning of November and its recovery in February, simultaneous to the high photochemical activity at mid-summer at Concordia, suggests the occurrence of a (photo-)chemical destruction of MSA taking place in submicron particles at that time. Under conditions encountered in the marine atmosphere, in contrast to its fast heterogeneous production, a significant (but slow) in cloud destruction of MSA is suspected to take place (Von Glasow and Crutzen, 2004; Barnes et al., 2006; Hoffmann et al., 2016). After its production, MSA present in air masses travelling inland Antarctica will encounter more oxidative conditions, specially when air masses remained for a few days over the high plateau and is thus characterized by high ozone mixing ratio (see discussions in Legrand et al. (2016))."**

Page 10, lines 11-14: I have some difficulty in following the logic here. Please clarify, especially what you mean by overestimation here and what is the actual reason for it. **As we stated, it is due to possible impact of pollution from the city of La Paz.**

Page 12: lines 5-13: The linear fits given in Table 4 should also be presented in Figure 11. **OK, we now report the regression lines in Figure 11 (corresponding to calculations reported in Table 4), as also required by the other reviewer.**

Besides correlations, how significant were the derived relations? **The standard error of the estimates of the slope and the y-intercept are presented in Table 4.**

A statement such as "the linear relationship . . . even less linear. . ." is meaningless/incorrect in a statistical sense and should be modified. **We agree and the text was modified as follows: "In fall, the relationship between nssSO$_4$ and MSA becomes even less linear than in spring......."**

Minor issues:
Page 3, line 12: I do not think that this is an acceptable way of citing other ACP papers (Legrand et al. this issue). **Yes, it is, since it is a companion paper.**

Page 5, line 15: weakness in abundance? Strange wording. **Ok we reworded this sentence as "We then focus discussions on the striking drop of the MSA to NssSO$_4$ ratio observed during mid-summer over inland Antarctica (Sect. 3.2)."**

section 3.2.2: should it be supermicron particles rather than micron particles? Page 8, line 8: . . .the higher. . .the higher. **OK Done.**
Please check out the grammar Page 9, line 4: des-appearance. Please check out the wording. **OK we corrected this.**

---

## Author Comment (AC2) · 20 Oct 2017

GENERAL : The paper presents measurements, results and analyses of sulfur aerosols at the Concordia station. The work is carefully done, it is a valuable paper for the interpretation of Antarctic aerosols and ice cores. As a highlight I would mention the interesting result of the interpretation of the MSA/nssSO4 and the photochemical destruction of MSA in summer. I can recommend publishing the paper in ACP, I only have minor revision suggestions.

*We first would like to thank the reviewer for its helpful comments (see our detailed answers below). We identify 10 bulk sulfate and MSA erroneous values (in January/March 2009 and 2010). They were corrected and all figures and Tables accordingly. These erroneous values however did not change the overall results and conclusions.*

DETAILED COMMENTS

The time series is fairly long – are there any statistically significant trends? Yes or no, it would be potentially important. **Yes we now addressed this point as follows " The aerosol record at Concordia now covers a decade. The long-term sulfate and MSA trends were examined by calculating the regression line slopes through annual and monthly mean values. No significant trend can be observed. For instance, a very weak annual increasing rate of 1.9 ± 5.6 ng m$^{-3}$ yr$^{-1}$ is calculated for sulfate in summer, however the regression line slope was found to be not statistically different from zero at the P > 95% confidence level."**

In the methods section: - sulfate might also come from the stations generator – could it? - was there any sector control? **Thank you for this comment and this point is now addressed in the text as follows: "The wind was occasionally blowing from the generator building of the Concordia station, disturbing measurements of atmospheric species like ozone (Legrand et al., 2016). Ozone measurements were also occasionally disturbed under very low wind speed conditions (< 2 m s$^{-1}$). The effect of such sporadic contamination of the station activities on the sulfate levels was here examined in the light of weekly denuder tubes sampling of acidic gases done at the site, as detailed by Legrand et al. (this issue) for HCl and HNO$_3$. Indeed, the denuder tubes sampling of acidic gases conducted at Concordia also document SO$_2$ by measuring sulfate on the extracts. After subtraction of a mean blank of sulfate of 1.5 ng m$^{-3}$ (i.e., 0.3 pptv of SO$_2$), the average mixing of SO$_2$ collected from January 2013 to April 2016 (170 samples) is of 0.7 ± 0.6 pptv. It is therefore unlikely that the station activities had emitted large enough amount of SO$_2$ to disturb the sulfate levels. "**

P4L8-11 "sulfate depletion relative to sodium with respect to the seawater composition .." there is the reference to the full paper but you could add a sentence or two as an explanation of the depletion here, too. **OK done as follows: "Examination of the size-segregated composition of aerosol present at Concordia indicates significant sulfate depletion relative to sodium with respect to the seawater composition from May to September (i.e. a k$_{SO4/Na}$ value of 0.16 ± 0.09 instead of**

**0.25 in seawater) (Legrand et al., this issue), resulting from the presence at the site of sea-salt aerosol emitted from both open ocean and sea-ice."**

Section 3.2.2 I am missing some comparison of HV and impactor data. I guess it has been done. A scatter plot with explanations would be nice. **Yes, we report the comparison in the figure below. We also add some sentences on the text: "Over the 2009 to 2011 years, aerosol was sampled on both bulk filter and impactor. A good agreement between the two data sets is found for sulfate as well as MSA (not shown). For sulfate, the relationship between the sum of concentrations observed on the impactor ($[SO_4]_{impactor}$) and the concentration observed on the bulk filter ($[SO_4]_{bulk}$) is $[SO_4]_{impactor} = 0.91 (\pm 0.08) * [SO_4]_{bulk}$ with $R^2 = 0.75$. For MSA the relationship is $[MSA]_{impactor} = 0.75 (\pm 0.06) * [MSA]_{bulk}$ with $R^2 = 0.8$. The slight difference between the two data sets is likely due to differences (up to a few days) in the sampling time intervals.»**

[Figure]

In Fig 4: there are the average size distributions of the respective seasons. How about showing there some range? **OK we change Figure 4 (a-d) showing the range for DC impactors.**

Also Becagli et al. (Atmos. Environ., 52, 98–108, 2012) showed size distributions measured at Dome C – make some comment on the main differences. **Yes but it is quite difficult to compare since Becagli and co (2012) used an impactor that samples submicron aerosol (above 0.4 micron) on two stages only (instead of 8 stages in our case above 0.028 micron) whereas most of sulfur biogenic mass is present between 0.1 and 1 micron. Nevertheless we have now referenced this study and results are reported in Table 2.**

P8,L6-8 "Impactor data corresponding to the March-November time period (Fig. 7) show that RMSA is very poorly related to the nssSO4 content (R2 of 0.01 and 0.06 for submicron and micron particles, respectively)". Fig 7 shows R vs MSA, not R vs nssSO4. I suggest adding subfigures where this is shown. **OK we agree and Figure 7 was modified accordingly.**

P10L9-10 "Assuming a sulfate concentration of 250 ng m-3 for the continental free troposphere of the southern hemisphere, and applying a dilution factor of 18 based on 210Pb data" Please explain how the dilution factor of 18 was obtained. Any uncertainty estimate for it? **OK we now specify in more details how these calculations were done: "Apart from marine biogenic emissions, sulfate present over Antarctica can also originate from southern hemisphere continents or the stratospheric reservoir. $^{210}$Pb data permit to derive an estimate of the contribution of sulfate long-range transported from continents by comparing the $^{210}$Pb concentrations at Concordia (27 µBq m$^{-3}$) after having corrected them from marine $^{222}$Rn exhalation (~ 15%, Weller et al. 2014) (i.e., 23 µBq m$^{-3}$) with those observed at Chacaltaya (407 µBq m$^{-3}$, Feely et al., 1988), a remote site located at 5220 m asl in Bolivia............ Assuming a sulfate concentration of 250 ng m$^{-3}$ for the continental free troposphere of the southern hemisphere, and applying a dilution factor of 18 based on $^{210}$Pb data (407 µBq m$^{-3}$ at Chacaltaya compared to 23 µBq m$^{-3}$ at Concordia), we calculate a mean sulfate concentration of 14 ng m$^{-3}$."**

P10L26-27 "Considering a mean sulfate mixing ratio of 0.3 ppbm for the lower stratosphere, we estimate that stratospheric-tropospheric exchange may account for 0.4 ng m$^{-3}$ of sulfate" Is 0.3 ppb = 0.4 ng/m$^{-3}$? **No, but may be the text was not clear enough that the value of 0.4 ng m$^{-3}$ was estimated by applying the dilution factor observed for $^{10}$Be to the 0.3 ppbm of sulphate present in the lower stratosphere. We have now reworded the sentence as follows: «Considering a mean sulfate mixing ratio of 0.3 ppbm for the lower stratosphere, and the typical dilution factor observed for $^{10}$Be between the lower stratosphere and the atmosphere at Concordia, we estimate that stratospheric-tropospheric exchange may account for 0.4 ng m$^{-3}$ of sulfate in winter at Concordia.**

Table 2 shows R in midsummer. March is not really midsummer any more. **Right and this was corrected in the Table caption as "in mid-summer and in February/March (if available).**

Figures with scatterplots: why don't you show the regressions there? **OK, we now report the regression lines in Figure 11 (corresponding to calculations reported in Table 4), as also required by the other reviewer. We don't do it for Figure 10 since that will overload the figure.**